# Selective Permeability of Volatile Organic Compounds in Candelilla Wax Edible Films

**DOI:** 10.3390/foods15020233

**Published:** 2026-01-09

**Authors:** Samuel Macario Padilla-Jiménez, Jose Manuel Oregel-Zamudio, Sergio Arias-Martínez, Jesús Rubén Torres-García, Ernesto Oregel-Zamudio

**Affiliations:** 1Instituto Politécnico Nacional, Centro Interdisciplinario de Investigación para el Desarrollo Integral Regional (CIIDIR), Unidad Michoacán, Jiquilpan 59510, Mexico; spadillaj1700@alumno.ipn.mx (S.M.P.-J.); joregelz2400@alumno.ipn.mx (J.M.O.-Z.); sariasm@ipn.mx (S.A.-M.); 2Escuela Nacional de Estudios Superiores, Unidad Morelia, Universidad Nacional Autónoma de México, Morelia 58190, Mexico; jtorresg@secihti.mx; 3Investigadores por México, Secretaría de Ciencia, Humanidades, Tecnología e Innovación, Mexico City 03940, Mexico

**Keywords:** candelilla wax, edible films, selective permeability, volatile organic compounds, HS-SPME/GC-MS

## Abstract

This study screens the permeability of volatile organic compounds (VOCs) through edible films made of candelilla wax and guar gum, offering new insights into their role as aroma and moisture barriers. Four formulations (0.2–0.4% wax, 0.4–0.8% gum, and 0.2–0.3% glycerol) were tested using a fractional factorial design. VOC fluxes (one ester, two aldehydes, two terpenes, and one lactone) were monitored via headspace solid-phase microextraction coupled to gas chromatography–mass spectrometry (HS-SPME/GC-MS) in a diffusion cell and modeled kinetically. Wax-rich matrices compacted the network, reducing initial VOC transmission by up to 60%, while glycerol fine-tuned micromobility and selectivity. The formulation containing 0.4% wax, 0.8% gum, and 0.2% glycerol minimized time-dependent flux acceleration and reduced the cumulative permeability of both polar (hexanal) and non-polar (limonene) markers by 80%. Aroma loss decreased across all blends, correlating with improved water vapor control. These results establish quantitative criteria for developing sustainable edible coatings that balance aroma retention, water-barrier performance, and mechanical flexibility.

## 1. Introduction

Loss of volatile aroma compounds is a direct determinant of sensory acceptance and is therefore a critical performance parameter that still lacks standardized metrics in edible films. Preserving food sensory integrity (aroma in particular) has become a priority for the industry on par with traditional requirements for safety and shelf life [1]. Modern consumers believe that the organoleptic profile should remain virtually unchanged from processing to consumption, a demand that has driven the development of active and selective packaging systems [2].

At the same time, regulatory and societal pressure to reduce the environmental impact of petro-based packaging has renewed interest in edible coatings formulated from biopolymers and natural waxes [3]. Among these, candelilla wax (obtained from *Euphorbia antisyphilitica*) stands out for its high hydrophobicity, elevated melting point, and regional availability attributes that flavor its incorporation into sustainable film-forming matrices [4,5,6]. Previous studies have shown that candelilla wax effectively reduces moisture loss and oxygen transfer [4,7]; however, the diffusion of volatile organic compounds (VOCs) through candelilla-based films remains poorly explored, even though these compounds govern the sensory acceptance of fresh and processed products [8,9,10,11].

From a transport standpoint, VOC permeation through edible films is commonly described by a coupled partitioning–diffusion mechanism, in which permeability reflects both (i) the effective affinity/sorption of each volatile in the film phases and (ii) the effective diffusivity through the polymer network [12,13]. Under practical conditions, relative humidity can modulate both contributions by plasticizing the hydrophilic fraction and shifting sorption equilibria, which may alter selectivity among VOCs with different polarity and phase affinity [14]. This framework is particularly relevant for multiphase hydrocolloid–wax systems, where wax domains can reduce water uptake, increase the effective hydrophobic fraction, and reorganize microstructural pathways, potentially changing tortuosity and phase availability for VOC transport. Within this class, candelilla wax is especially attractive because its high hydrophobicity and relatively high melting point enable robust hydrophobic structuring in composite films, and its applicability has been demonstrated in edible coatings for postharvest preservation [15,16], with recent reviews highlighting its suitability for edible films/coatings and sustainable food-related formulations [6].

Consistent with this framework, the literature indicates that aroma permeability not only depends on the lipid fraction but also on the polysaccharide phase and the plasticizer that imparts network flexibility [17,18]. Guar gum is commonly used as a film-forming agent owing to its high viscosity and ability to create continuous networks, whereas glycerol acts as a plasticizer, reducing brittleness [19,20,21]. Nevertheless, the three-dimensional interaction among wax, polysaccharide, and glycerol (and its combined impact on VOCs selectivity) has not been addressed systematically.

Against this backdrop, the present work tests two hypotheses: (i) that the microstructure arising from varying the wax–gum–glycerol ratio modulates, in a differentiated manner, the diffusion of VOCs representative of distinct chemical classes; (ii) that kinetic characterization (via linear and quadratic models) discriminates between purely steric effects and plasticization phenomena induced by volatile sorption. To test these hypotheses, a fractional factorial experiment generated four formulations with contrasting levels of candelilla wax, guar gum, and glycerol. The resulting films were evaluated by gravimetric techniques (water-barrier properties) and HS-SPME coupled with GC-MS, which enabled the real-time monitoring of six key VOCs: one ester, two aldehydes, two terpenes, and one lactone. Kinetic data were fitted to first- and second-order models to obtain intrinsic transmission rates and permeability coefficients, providing a comprehensive view of barrier performance. The study not only fills the knowledge gap on the aromatic selectivity of candelilla wax but also supplies formulation criteria for designing edible coatings capable of preserving or blocking specific volatiles aligned with modern sustainability and sensory quality demands in the food chain. Importantly, establishing these intrinsic VOC transport parameters under controlled conditions provides the necessary foundation for subsequent, matrix-specific coating trials in real foods, where respiration, exudates, and film–food interactions must be addressed through dedicated experimental designs.

## 2. Materials and Methods

### 2.1. Reagents and Reference Standards

Food-grade guar gum (≥98%; CAS 9000-30-0; Sigma-Aldrich, St. Louis, MO, USA), refined candelilla wax (≥99%; Multiceras S.A. de C.V., Monterrey, Mexico), and anhydrous glycerol (≥99.5%; CAS 56-81-5; Sigma-Aldrich, St. Louis, MO, USA) were used as film-forming components. All volatile organic compounds (VOCs) were purchased from Sigma-Aldrich (St. Louis, MO, USA): ethyl butyrate (≥98%, FCC, FG, CAS 105-54-4), hexanal (≥97%, FCC, FG, CAS 66-25-1), trans-2-hexenal (≥95%, FCC, FG, CAS 6728-26-3), (R)-(+)-limonene (97%, CAS 5989-27-5), (−)-linalool (natural, ≥95%, FG, CAS 126-91-0), and γ-decalactone (≥98%, FCC, FG, CAS 706-14-9). Each analyte was separately dissolved in HPLC-grade ethanol (LiChrosolv^®^, Merck, Darmstadt, Germany) to 1 mg mL^−1^; stock solutions were filtered through 0.22 µm PTFE, stored in amber vials with PTFE/silicone septa, and kept at 4 ± 1 °C for up to six weeks. n-Decyl ethanoate (ethyl n-decanoate, ≥98%; CAS 110-38-3; Sigma-Aldrich) was added as an internal standard at 50 ng mL^−1^ to compensate for instrumental variability; the deviation of its chromatographic area between consecutive injections did not exceed 3%. Saturated salts (ACS grade, Sigma-Aldrich) of MgCl_2_ (33% RH) and NaNO_3_ (67% RH) were prepared following ASTM E104 [22] to control relative humidity. Anhydrous NaCl (≥99%), regenerated at 200 °C for 3 h, served as sorbent in gravimetric water vapor tests. All water used was produced by a Milli-Q IQ 7000 system (18.2 MΩ cm; Merck Millipore, Molsheim, France). Prior to use, all glassware was silanized (Sigmacote^®^, Sigma-Aldrich) and conditioned at 120 °C for 2 h to minimize VOC adsorption.

### 2.2. Fractional Factorial Design and Formulation

A 2^3−1^ fractional factorial design (generator ABC = +1, resolution III) was used as a screening approach to estimate the direction and relative magnitude of the main effects of guar gum, candelilla wax, and glycerol at two levels (0.4–0.8% gum; 0.2–0.4% wax; and 0.2–0.3% glycerol). The experimental matrix comprised four formulations, EFA (0.4% gum, 0.2% wax, and 0.2% glycerol); EFB (0.4%, 0.4%, and 0.3%); EFC (0.8%, 0.2%, and 0.3%); and EFD (0.8%, 0.4%, and 0.2%), whose preparation order was randomized. To ensure repeatability, three independent batches of each formulation were produced; batch-to-batch coefficients of variation remained below 5% for thickness and below 7% for barrier properties. All solids were weighed on an analytical balance (MS105, Mettler-Toledo, Greifensee, Switzerland). Ingredient traceability was recorded by internal lot code and opening date. By design, main effects may be aliased with two-factor interactions; therefore, interpretations are confined to main-effect trends.

### 2.3. Film Preparation

An aqueous dispersion was prepared by heating 100 mL of distilled water to 80 ± 1 °C on a hot-plate stirrer (C-MAG HS 7, IKA-Werke, Staufen, Germany); temperature was monitored with a digital thermometer. Guar gum was fed at 0.5 g min^−1^ while homogenizing at 18,000 rpm with a high-speed homogenizer. After 5 min of hydration, pre-melted and filtered candelilla wax (85 °C, 150 µm mesh) was slowly added. Once the system cooled to 50 °C, glycerol was introduced, and homogenization continued for 5 min. The emulsion was degassed under dynamic vacuum (RV5 rotary pump, Edwards Vacuum, Crawley, UK), poured into pre-leveled 90 mm Petri dishes on an anti-vibration platform, and the liquid mass deposited (25 ± 0.05 g) was verified with a precision balance. Drying was carried out in a drying oven (UF55, Memmert GmbH, Schwabach, Germany) at 30 ± 1 °C and 40% internal RH; mass was checked every 3 h; and drying was deemed complete when the change was <1 mg h^−1^. After 24 h, the films were detached with a PTFE spatula and stored in low-permeability laminated pouches.

### 2.4. Physicochemical Determinations

#### 2.4.1. Thickness, Volume, Mass, and Density

Sample perimeters were measured with a digital Vernier caliper (model 500-196-30; Mitutoyo, Kawasaki, Japan). Thickness was measured at ten positions per specimen (eight around the perimeter at 45° and two central points) using a digital Vernier caliper; the mean value was used for calculations. Assuming planar geometry, volume was calculated from length, width, and thickness, after which each sample was weighed on an analytical balance (MS105, Mettler-Toledo, Greifensee, Switzerland). Film density (ρ) was obtained from Equation (1):
(1)$$\rho =\frac{m}{v}$$ where
m is the sample mass (g) and
v its volume (cm^3^). Five specimens (2 cm^2^) per formulation were analyzed.

#### 2.4.2. Moisture Content

Moisture was determined with a thermobalance (MA160, Sartorius AG, Göttingen, Germany). Five independent samples (2 cm^2^) per formulation were placed in pre-tared aluminum pans and dried at 110 ± 2 °C to constant weight. Moisture (%) was calculated with Equation (2):
(2)$$M\left(\%\right)=\frac{w1-w2}{w1}\times 100$$ where
w1 and
w2 are the initial and final weights, respectively. Each measurement was triplicated; dried samples were cooled in a desiccator prior to final weighing.

#### 2.4.3. Solubility in Water

Samples (2 cm^2^) were pre-dried at 80 ± 1 °C for 24 h to obtain the initial dry mass (dw1). They were then immersed in 80 mL of distilled water at 25 ± 2 °C under magnetic stirring (150 rpm) for 10 min, removed, and dried again at 80 ± 1 °C to constant mass (dw2). Solubility (%) was calculated with Equation (3). At least three replicates per formulation were performed.
(3)$$S\left(\%\right)=\frac{dw1-dw2}{dw1}\times 100$$

### 2.5. Water Vapor Permeability

#### 2.5.1. Water Vapor Transmission Rate (WVTR)

WVTR was measured gravimetrically using a 100 mL glass cell containing 5 g of anhydrous CaCl_2_, previously equilibrated in a desiccator. Each film was sealed over the cell mouth, the initial mass was recorded, and the assembly was placed in a desiccator maintained at 100% RH (distilled water). Mass gain due to water sorption was recorded hourly for ≥5 h until a linear increase was obtained. The slope divided by the exposed film area yielded WVTR (Equation (4)):
(4)WVTR=Slope×Atrasf where
Slope e is the rate of mass change (g s^−1^), and
Atrasf is the exposed area (m^2^); here, a 90 mm diameter base corresponds to 0.00636 m^2^.

#### 2.5.2. Water Vapor Permeability (WVP)

WVP was calculated by multiplying WVTR by film thickness
${(X}_{film}$) and dividing by the water vapor pressure gradient at 25 °C (Equation (5)):
(5)$$WVP=\frac{WVTR\times X_{film}}{∆P}$$ where
$X_{film}$ is the film thickness (m), and
∆P is the vapor pressure difference at 25 °C (2337 Pa).

### 2.6. Permeability to Volatile Organic Compounds

The permeability of the edible films to VOCs was quantified with a hermetically sealed, two-chamber glass diffusion cell (Figure 1). Each film was clamped between the chambers, defining an exposed area of 0.00636 m^2^. The donor chamber contained 100 mL of a reference solution (10 ng kg^−1^ each of ethyl butyrate, hexanal, trans-2-hexenal, (R)-(+)-limonene, linalool, and γ-decalactone) and was equilibrated to 33% relative humidity (RH) with MgCl_2_. The receiver chamber was held at 67% RH with NaNO_3_ and continuously purged with chromatographic-grade helium to remove desorbed volatiles. The entire assembly was maintained at 25 ± 1 °C.

Headspace sampling was performed every 2 h for 12 h. An HS-SPME fiber (50/30 µm) was exposed in the receiver headspace for 5 min and thermally desorbed in the GC-MS injector for quantification. For each film–compound pair, the cumulative mass per unit area in the receiver, C (t) (ng m^−2^), was fitted by least squares to both a linear model (Equation (6)) and a quadratic model (Equation (7)).
(6)$$\mathrm{Linear}\;\mathrm{model}:\;C\left(t\right)=\beta_{0}+\beta_{1}t$$
(7)$$\mathrm{Quadratic}\;\mathrm{model}:\;C\left(t\right)=\beta_{0}+\beta_{1}t+{\beta_{2}t}^{2}$$

#### 2.6.1. Volatile Organic Compound Transmission Rate (VOCTR)

The instantaneous volatile transmission rate, VOCTR(t), was calculated as the derivative of C (t) divided by the exposed area (*A*) (Equation (8), linear model; Equation (9), quadratic model).
(8)$${VOCTR}_{linear}=\frac{\beta_{1}}{A}$$
(9)$${VOCTR}_{quadratic}=\frac{\beta_{1}+2\beta_{2}t}{A}$$

#### 2.6.2. Volatile Organic Compound Permeability Coefficient (VOCP)

To account for film thickness
${(X}_{film}$) and for the vapor pressure gradient Δ*P* (795 Pa, corresponding to 67% vs. 33% RH at 25 °C), the volatile permeability coefficient was calculated as follows: linear model (Equation (10)) and quadratic model (Equation (11)).

Linear model:
(10)$${VOCP}_{linear}=\frac{{VOCTR}_{linear}\times X_{film}}{∆P}=\frac{X_{film}}{A∆P}\beta_{1}\;$$

Quadratic model:
(11)$${VOCP}_{quadratic\left[0,h\right]=}\frac{X_{film}}{∆P}\;\int_{0}^{h}\left({VOCP}_{quadratic}(t)\right)dt=\frac{X_{film}}{A∆P}(\beta_{1}+\beta_{2}h)$$

All permeability assays were performed in triplicate (n = 3).

### 2.7. Headspace Solid-Phase Microextraction (HS-SPME)

VOCs permeating the edible films were captured by HS-SPME using a DVB/CAR/PDMS-coated fiber (50/30 µm; Supelco^®^, Bellefonte, PA, USA). The fiber was pre-conditioned in the GC injector at 230 ± 1 °C for 15 min to eliminate residual contaminants and stabilize the coating. For each analysis, the fiber was placed in the receiver headspace (Figure 1) for 30 min at 25 ± 1 °C. This exposure time optimized VOC adsorption with respect to volatility and fiber capacity. The fiber was then immediately transferred to the GC injector, and desorption was carried out at 230 ± 1 °C for approximately 5 min. The procedure was repeated systematically after each sampling interval.

### 2.8. Gas Chromatography–Mass Spectrometry (GC-MS)

Separation and detection of VOCs were performed on a Clarus 680 GC system coupled to a Clarus SQ8T MS (PerkinElmer, Shelton, CT, USA) fitted with an Elite-5 capillary column (30 m × 0.32 mm × 0.25 µm). Helium (99.999% purity), at a constant flow of 1.00 mL min^−1^, served as the carrier gas. The oven program started at 30 °C (2 min), ramped at 9 °C min^−1^ to 140 °C, and held for 5 min. The injector, operated in splitless mode, was maintained at 230 ± 1 °C to ensure complete vaporization of analytes desorbed from the HS-SPME fiber. The mass spectrometer ran under electron-impact ionization (70 eV); source and transfer-line temperatures were stabilized at 250 °C and 230 °C, respectively. Screening/ID was first optimized in full scan (m/z 30–400) to confirm library matches; quantification was performed in Selected Ion Recording (SIM/SIR) monitoring target/qualifier ions for each VOC.

### 2.9. Identification and Quantification of VOCs

Peak assignment was based on matching retention times with authentic standards analyzed under identical chromatographic conditions; identity was confirmed by comparing mass spectra with the NIST/EPA/NIH v 2.2 library [23], accepting only matches with similarity indices ≥ 90%. Quantification relied on external calibration curves (0.10–10.0 ng µL^−1^, five levels) prepared in aqueous matrix and subjected to the same desorption protocol. Weighted fitting (1/x) of the peak area ratio to the internal standard (ethyl n-decanoate, 2 ng µL^−1^) yielded coefficients of determination ≥ 0.998. Sample concentrations were obtained by direct interpolation and internal normalization to compensate for instrumental variability. Quality control included double fiber blanks and verification standards every ten injections; relative deviation of the internal standard area remained ≤ 5% and in-sequence precision was (duplicates) ≤ 5%.

### 2.10. Analytical Recovery and Detection Limits

Extraction efficiency was evaluated with standards spiked into aqueous matrix and processed under identical HS-SPME/GC-MS conditions. Mean recoveries (n = 3) ranged from 82 to 87% for ethyl butyrate, 75–83% for hexanal, 93–96% for trans-2-hexenal, 71–75% for (R)-(+)-limonene, 80–87% for linalool, and 89–93% for γ-decalactone, variations attributable to differences in volatility and polarity among analytes and their differential affinity for the DVB/CAR/PDMS coating. Detection limits, calculated as the minimum concentration producing a signal three times the blank noise, ranged from 0.01 to 0.09 ng g^−1^.

### 2.11. Scanning Electron Microscopy (SEM)

Surface microstructure was examined by SEM using a JEOL JSM-6610 (JEOL Ltd., Tokyo, Japan) operated in high vacuum, 15 kV accelerating voltage, 10 mm working distance, spot size 54, and SEI detection. Rectangular fragments (~5 × 5 mm) were mounted on aluminum stubs with conductive carbon tape, selecting non-overlapping, defect-free fields. Images were acquired at ×500 with a 50 μm scale bar; instrument metadata (mode, kV, WD, and spot) were recorded automatically. To minimize charging, spot size and dwell time were adjusted; when necessary, a very thin conductive coating was applied prior to acquisition. Post-processing was limited to uniform linear brightness/contrast adjustments; no geometric transforms or non-linear filtering were used. The micrographs are derived from equivalent batches and are reported qualitatively; study conclusions rest on the transport data presented elsewhere.

### 2.12. Statistical Analysis

All calculations were performed in R 4.4.2 [24]. A multifactor ANOVA corresponding to a 2^3−1^ fractional factorial design was first applied to the responses of film thickness, apparent density, moisture content, solubility, WVTR, and WVP. The model included only the main effects of wax, gum, and glycerol; two-factor interactions were not estimated or interpreted due to aliasing inherent to Resolution-III designs. Statistically significant main-effect terms (*p* ≤ 0.05) were further analyzed using Tukey’s HSD test (α = 0.05).

The volatile transmission rate and permeability coefficients derived from the linear model (VOCTR_linear_ and VOCP_linear_) were estimated from the slope parameter (β), normalized by film thickness and vapor pressure gradient. These parameters were then analyzed using two-way ANOVA (4 Treatments × 6 Metabolites, N = 30), followed by Sidak-adjusted multiple comparisons to compare main-effect trends across formulations within each analyte; interaction patterns are not inferred within this screening design. Assumptions of residual normality and homoscedasticity were tested using the Shapiro–Wilk and Levene’s tests, respectively.

In parallel, each diffusion profile (formulation × metabolite, 0–12 h) was also fitted to a quadratic model by ordinary least squares. The corresponding instantaneous rates were likewise normalized by film thickness and vapor pressure gradient to compute VOCTR_quadratic_ and VOCP_quadratic_. These quadratic parameters were structured into a three-dimensional matrix (6 metabolites × 4 formulations × 6 time intervals), which was z-standardized and visualized as a heat map. Hierarchical clustering was applied to both rows and columns using Euclidean distance and average linkage, while cluster robustness was evaluated through 1000 multiscale bootstrap replicates.

## 3. Results

### 3.1. Physicochemical Characterization of the Films

The six panels in Figure 2 show how the gum–wax–glycerol ratio reshapes film micro-architecture without compromising process reproducibility. Thickness remained tightly confined to 0.05–0.08 mm for all formulations (Figure 2a), and the near-complete overlap of the box plots confirms that the casting–drying protocol decoupled composition from geometry.

Density (Figure 2b) provides the first clear contrast: formulation EFB, densified by wax and glycerol, exhibits the highest values (12 mg cm^−3^), while the wax-poor EFA clusters around 8 mg cm^−3^. This upward shift signals lipid-induced compaction, an effect echoed in other variables. Compaction also modulates internal hydration: in Figure 2c, the blend with the greatest polysaccharide fraction (EFC) reaches 16% moisture two percentage points above EFA. The narrow scatter of points corroborates high experimental repeatability and the hygroscopic capacity of guar gum. The trend reverses for solubility (Figure 2d): EFC, the most hydrated matrix, loses 85% of its mass in the immersion test, whereas EFA scarcely exceeds 70%. Shorter whiskers suggest that solubility is an intrinsic network attribute, less sensitive to minor thickness variations.

Water-barrier behavior is more nuanced. Despite its hydrophobicity, EFA records the highest WVTR (0.39 g s^−1^ m^−2^) because its low density creates preferential diffusion channels (Figure 2e). In contrast, EFB drops below 0.30 g s^−1^ m^−2^ thanks to wax-induced densification and the observed reduction in effective pathways under the tested conditions. Yet, once normalized for thickness and the vapor pressure gradient (Figure 2f), the curves converge: intrinsic diffusive resistance is comparable after removing geometric effects.

Overall, Figure 2 suggests that wax, gum, and glycerol act as opposing modulators within this dataset: Wax compacts and protects against water, but without a sufficient plasticizer, it can create heterogeneities; gum increases water uptake and solubility; glycerol acts as a fine tuner, which may reduce free-volume pathways or soften the network, depending on its level.

### 3.2. Initial Linear Kinetics of Volatile Diffusion

During the first 12 h, accumulation curves for the six VOCs (Figure 3) display linear behavior (R^2^ > 0.85; *p* ≤ 0.05), indicating a virtually constant activity gradient. The tight point scatter confirms film homogeneity and HS-SPME/GC-MS reproducibility.

For every analyte, EFA is the least dense, wax-poorest matrix, showing steeper slopes, especially for γ-decalactone and limonene, tripling those of EFB or EFD. The contrast shrinks but persists for lower-mass aldehydes (hexanal, trans-2-hexenal) and for ethyl butyrate. Differences emerge between EFB and EFC: although EFB contains more wax, its aldehyde rates exceed EFC’s because additional glycerol increases free volume and the diffusion of polar volatiles. EFC, rich in gum, densifies the hydrophilic phase and compensates for its lower lipid fraction. EFD (maximum wax, minimum glycerol) shows the slowest kinetics; ethyl butyrate is almost completely blocked. In summary, wax raises tortuosity; gum retains water and strengthens the hydrophilic continuum; and glycerol modulates segmental mobility according to analyte polarity.

### 3.3. Transition to the Quadratic Regime

After 12 h, several curves deviate from linearity (Figure 4). Quadratic fitting improves the R^2^ by 6–18 pp (*p* ≤ 0.05). The transition is most pronounced in low-density films: in EFA, γ-decalactone and limonene accelerate after 8 h, consistent with non-stationary transport (e.g., sorption-softening). Glycerol moderates the effect in EFB (the β_2_ (quadratic coefficient) decreased by ~40%). In EFC, the curvature is minimal except for linalool; in EFD, the quadratic terms are negligible. The magnitude of the quadratic coefficient was reported descriptively as having a higher curvature in low-density matrices, while wax-enriched blends showed attenuated curvature. We refrain from mechanistic assignments within this screening design. The phenomenon is analyte-dependent: more hydrophobic compounds show greater curvature; low-mass aldehydes maintain nearly constant slopes. Thus, the wax–gum–glycerol ratio controls not only the initial speed but also the temporal stability, which is critical for prolonged performance. Curvature may also arise from non-steady diffusion, saturation, or experimental artifacts; therefore, the linear/quadratic fits are interpreted as empirical descriptors of temporal trends rather than mechanistic proof.

### 3.4. Linear Transport Parameters

Quantitative parameters (Figure 5) corroborate kinetic trends: EFA yields the highest VOCTR_linear_, while blends enriched in wax and/or gum drastically reduce fluxes. γ-decalactone shows the greatest rate, followed by trans-2-hexenal. EFA exceeds EFB by 30% and EFC/EFD by 40%. After thickness normalization (VOCP_linear_, Figure 6), EFA still ranks highest, although its advantage falls to 15% for several analytes, confirming that open microstructure rather than geometry dominates permeability. Linalool rises from EFA to EFD despite similar thicknesses, exploring polar domains in the dense EFD matrix. The strictly hydrophobic limonene remains penalized in EFD, consistent with wax selectivity. Replacing 0.2% gum with wax (EFA to EFB) cuts the rate by 60%; glycerol fine-tunes selectivity according to polarity.

### 3.5. Integrated Quadratic Parameters

Heat maps of instantaneous rates (Figure 7) reveal that the diffusive regime ceases to be stationary only in specific “formulation × analyte” pairs. In EFA, linalool quintupled its rate between 2 h and 12 h (>50 ng h^−1^ m^−2^); γ-decalactone and trans-2-hexenal show similar patterns. Hierarchical clustering separates three groups: (i) EFA records with apolar compounds (highest permeability), (ii) a middle group with all EFB/EFC data and moderate rates, and (iii) EFD, whose quadratic term is insignificant. Wax-induced densification and lower glycerol thus stabilize the barrier against self-acceleration.

Accumulated permeability curves (Figure 8) magnify the differences: linalool–EFA reaches 2900 ng m^−2^; γ-decalactone and trans-2-hexenal exceed 2500 ng m^−2^. Equivalent pairs in EFD barely surpass 600 ng m^−2^; ethyl butyrate–EFD remains virtually unchanged, confirming the blockage. The horizontal dendrogram organizes records by polarity: aldehydes and the ester cluster centrally; terpenes and the lactone occupy the extremes, underscoring that wax effectiveness strongly depends on analyte solubility in the lipid phase.

Overall, initial density correlates with the linear slope β, while β_2_ captures non-stationary behavior in specific formulation–analyte pairs. Low wax and high residual moisture facilitate sorption-induced expansion and raise permeability; wax enrichment confers rigidity and suppresses temporal amplification. Adjusting the wax–glycerol ratio is therefore essential not only for the initial barrier profile but also for kinetic stability during real storage.

### 3.6. Surface Microstructure

Representative surface micrographs (SEI, 15 kV, WD 10 mm, spot 54, ×500, and 50 μm bar) display a globular topography with a multimodal distribution of domains. Semi-spherical features of ~10–35 μm are superimposed on finer populations (~2–8 μm) and a sub-micrometric granulation that carpets the background; no macroscopic open pores or discontinuities are evident in the fields analyzed. Across the four panels, domain areal density and coalescence vary: (Figure 9a) shows the highest concentration and local coalescence of large domes; (Figure 9b) shows a more homogeneous distribution with moderate relief; (Figure 9c) shows a central agglomerate of fine particles on a clearer background; and (Figure 9d) shows medium domes intermixed with slightly elongated features. These observations are qualitative and provided as visual support to the instrumental barrier results.

## 4. Discussion

Characterizing the permeability of volatile organic compounds (VOCs) must rank alongside standard physicochemical assessments of edible films, because aroma retention directly governs consumer acceptance. The primary aim of this work was to quantify VOC transmission through candelilla wax films and to elucidate how simultaneously adjusting guar gum and glycerol modulates barrier performance. This objective addresses the well-documented need to replace petro-plastic packaging with sustainable coatings capable of conserving foods’ aromatic fraction, an aspect scarcely explored in wax-based matrices [25,26]. Our strategy, a factorial design combined with HS-SPME extraction and linear quadratic kinetic modeling, differs from prior studies that focused almost exclusively on water or oxygen vapor fluxes [27,28,29,30].

Beyond underscoring sustainability, it is useful to benchmark aroma-barrier magnitudes against contemporary bio-based films. Recent reviews report that incorporating lipids (waxes, oils) into hydrophilic matrices typically lowers water- and aroma-permeation by creating discontinuous, low-polarity pathways; however, most datasets still focus on H_2_O/O_2_ rather than volatiles (VOCs) [17,31]. Our use of HS-SPME with kinetic fitting addresses this gap and is methodologically consistent with current recommendations for volatile transport in edible films. As an external reference point for aroma transport, multilayers based on PHBV or PET show limonene permeance on the order of 5.8 × 10^−10^ and 4.9 × 10^−11^ kg m^−2^ Pa^−1^ s^−1^, respectively, at 23 °C/50% RH [32]. These values contextualize the barrier levels that edible coatings must approach to effectively retain high-volatility terpenes.

The six VOCs targeted in this work are directly relevant to real food matrices and flavor outcomes. Ethyl butyrate is a key fruity ester in strawberry and apple products (juices, purées, and flavored dairy), where its retention under storage governs top-note intensity. Hexanal is a well-established marker of lipid oxidation and rancidity in lipid-rich systems (e.g., fried snacks, meats), so lowering its permeability is consistent with delaying off-flavor development. Trans-2-Hexenal contributes “green/leafy” notes characteristic of fresh apple and other produce, making its control pertinent to preserving fresh-like profiles in minimally processed fruits. In citrus matrices, (R)-(+)-limonene and (−)-linalool are major terpenes structuring the orange/mandarin aroma; reducing their loss supports the persistence of citrus top notes in beverages and confectionery. γ-decalactone imparts the “peach-like” character in stone-fruit derivatives and related dairy/beverage applications, so reduced permeability translates into better retention of these impact notes. Taken together, these analytes anchor the permeability measurements to application-oriented scenarios; the present results constitute a preliminary kinetic characterization intended to establish transport baselines prior to applied validation in specific food systems. Importantly, the six VOCs used in this work constitute a representative (non-exhaustive) marker set spanning key aroma-related chemical families and polarity/phase-affinity differences, enabling the controlled comparative screening of film VOC transport within a single HS-SPME/GC-MS protocol. Extension to acids, amines, sulfur volatiles, and broader food-specific volatilome profiling requires dedicated analytical approaches (e.g., derivatization and pH-controlled designs) and is proposed as future work.

The analyte set also enables mechanistic contrasts using solubility–diffusivity arguments. Terpenes such as limonene and linalool exhibit a high affinity for hydrophobic phases and are known plasticizers of polyesters, which increases their apparent diffusivity in rubbery or partially plasticized domains; reported limonene uptake in PHAs can exceed 10 wt% and accompanies a measurable rise in permeance [32]. Conversely, smaller, more polar aldehydes (hexanal, trans-2-hexenal) interact preferentially with hydrated polysaccharide regions, so their transport is more sensitive to matrix moisture and plasticizer content. These dual trends are consistent with current edible film transport frameworks [33].

Physicochemical results confirmed that moisture content is chiefly dictated by the polysaccharide-to-lipid balance. Formulation EFC, with the highest guar gum fraction, retained about 16% water, indicating a more hydrophilic network and, therefore, greater swelling propensity [34,35]. Works on chitosan- and starch-based films report the same link between moisture and permeability to polar solutes [36,37]; in our study, this translated into steeper aldehyde slopes in EFC, suggesting that residual water acts as an internal plasticizer that eases diffusion. The moisture–permeability linkage in our formulations aligns quantitatively with recent observations: guar-rich, water-swollen networks raise fractional free volume for polar penetrants, increasing their effective diffusion coefficients [38]. Guar gum is also intrinsically hygroscopic, promoting water uptake at ambient RH, which further plasticizes hydrophilic phases [39]. In polysaccharide films, glycerol exacerbates this effect by lowering Tg and enlarging micro-mobility, typically elevating WVP and solute permeability at a fixed RH [40]. The steeper aldehyde slopes we recorded for EFC are therefore consistent with a moisture-plasticization mechanism predicted by free-volume theory and recent hygroscopicity data.

Density emerged as an inverse proxy for permeability. EFB, densified by wax and glycerol, exhibited the highest density and the lowest transmission rates for most analytes. Similar findings have been described in protein-based biopolymers, where apparent densification reduces free volume and, consequently, the diffusion coefficient [41]. The 0.4% wax/0.3% glycerol blend offered the best compromise between density and flexibility without sacrificing mechanical integrity. Using density as a proxy for barrier performance is consistent with diffusion models in glassy/rubbery polymers, where permeability P = D × S falls as the fractional free volume decreases [38]. Practically, densification by wax and moderate glycerol lowers D by reducing accessible micro-voids while maintaining enough flexibility to avoid microcracking that would otherwise short-circuit the barrier. This rationale mirrors quantitative improvements seen in dense bio-polyester or wax-reinforced multilayers, where limonene permeance shifts downward by roughly one order of magnitude as the continuous phase becomes more compact and hydrophobic [32].

Aqueous solubility provided complementary insight. The most soluble films (again EFC) would be less suitable for foods with high surface moisture because the matrix would erode and lose aroma-retention capacity [42]. Adding wax lowered solubility by more than 15 percentage points, strengthening the barrier against hydrophilic compounds, as reported for blends with microcrystalline waxes [43]. The solubility decrease we observed upon adding wax agrees with reports that plant wax domains reduce matrix wettability and suppress aqueous erosion. In candelilla- or beeswax-containing films, this manifests as 10–70% reductions in WVP depending on the wax fraction, crystallinity, and dispersion scale [31,44]. From a design standpoint, minimizing film solubility is essential when targeting aroma retention in foods with wet surfaces, because dissolution not only thins the barrier but also enhances volatile partitioning to the headspace.

Regarding water vapor, wax-rich formulations showed the lowest WVTR and WVP values, consistent with lipid-dispersion models in which the lipid phase seals the hydrophilic continuum [17,45]. Studies using beeswax or carnauba wax describe comparable reductions, though absolute values depend on lipid crystallinity and domain distribution [46,47]. This effect is pivotal for preserving the texture of thin-skinned fruits and bakery products. Recent quantitative studies on wax coatings corroborate our WVTR/WVP trends: coating paper with beeswax lowered WVTR by ~77% vs. uncoated paper at similar test conditions, with performance strongly tied to wax layer continuity and crystallinity [48]. Complementary work on renewable nanostructured multilayers shows that achieving a continuous wax layer is a key determinant of high water vapor barriers, particularly under humid conditions [49]. Given candelilla’s high hydrocarbon fraction, our wax-rich formulations likely operate by the same sealing mechanism, with absolute values modulated by domain size and RH.

VOC kinetics revealed two regimes. Linear behavior predominated during the first 12 h, but positive curvature appeared in less-dense formulations, attributable to progressive plasticization [50]. Analogous phenomena have been reported in coatings containing essential oils, where terpene sorption softens the matrix and raises the diffusion coefficient [51,52]. In our case, linalool and limonene triggered this acceleration, whereas hexanal and ethyl butyrate scarcely altered the slope. The emergence of a positive curvature in VOC uptake is consistent with penetrant-induced plasticization: as terpenes sorb, they reduce local Tg and increase segmental mobility, raising D over time [53]. This time-dependent acceleration is expected to be strongest for highly soluble terpenes (limonene, linalool) in hydrophobic domains and weakest for aldehydes with a lower affinity for wax-rich regions. Capturing this behavior with a linear-then-quadratic (or dual-mode/Fickian-plus) model is therefore mechanistically justified and aligns with the polymer-transport theory used in contemporary barrier analyses [33].

Linear rates clearly distinguished blends: polar aldehydes diffused more readily in open matrices, whereas terpenes were effectively contained in dense, wax-laden films. Wax forms hydrophobic crystalline domains that increase tortuosity and present low polarity [54]; gum, by retaining water, raises local viscosity and defines effective channel size [55]; and glycerol fine-tunes micro-mobility, sealing or opening micro-voids according to its concentration [56]. This triad enables the simultaneous tuning of water and aroma barriers key to tailor-made formulations. Microstructurally, our “wax–gum–glycerol” triad maps onto diffusion–solubility control knobs: wax increases tortuosity and reduces the solubility of polar penetrants; guar governs water uptake and local viscosity; and glycerol tunes the free volume. In dual-mode terms, wax lowers the Henry’s law constant S for polar species and restricts the Langmuir capacity by filling micro-voids; glycerol shifts the diffusional prefactor through Tg depression, with a non-monotone optimum that we empirically locate near 0.2–0.3% (*w*/*w*) [33,40]. This decomposition explains the sharper discrimination we observed between aldehydes (favored by hydrated channels) and terpenes (favored by hydrophobic domains).

These microstructural arguments are qualitatively supported by SEM observations. The heterogeneous, multimodal globular morphology typical of hydrocolloid–lipid composites is consistent with a phase-separated structure in which wax-rich domains are dispersed within a continuous polysaccharide matrix. Within a coupled partitioning–diffusion framework, increasing the density and/or coalescence of lipid domains is expected to increase geometric tortuosity and disrupt pathway continuity, thereby reducing effective diffusivity for penetrants whose transport is dominated by the hydrophilic continuum [12,14]. Conversely, hydration of the polysaccharide phase promotes humidity-driven plasticization, increasing segmental mobility and fractional free volume and thus raising effective diffusivity, particularly for more polar VOCs that preferentially partition into hydrated regions [57]. Together, these features align with the formulation-dependent VOCTR/VOCP trends and with the time-dependent deviations from linearity captured by the quadratic fit in selected formulation–analyte pairs [13]. It should be emphasized that SEM is used here as qualitative evidence of phase distribution/coalescence rather than a direct quantification of porosity or free volume; complementary quantitative analyses under controlled humidity would further parameterize transport pathways and are proposed as future work [57]. Comparison of WVP and VOCP indicated that the barrier is more effective against polar than apolar molecules; optimization must therefore match each food’s aroma profile. Similar contrasts have been reported for starch films reinforced with oils, where water resistance increases without reducing terpene loss [58,59,60]. The aroma/water selectivity observed here mirrors recent biopolymer reports. For instance, at 23 °C/50% RH, limonene permeance spans ~2.2 × 10^−9^ (paper) → 5.8 × 10^−10^ (PHBV, 10 µm) → 4.9 × 10^−11^ kg m^−2^ Pa^−1^ s^−1^ (PET multilayer), emphasizing how hydrophobic continuity and lower free volume suppress terpene loss [32]. Our wax-laden blends behaved analogously strongly, impeding terpenes while leaving polar aldehydes comparatively less affected, implicating solubility selectivity as the dominant lever beyond tortuosity.

Mechanistically, VOCs diffusion is governed by steric exclusion (controlled by density) and differential solubility (affinity for the continuous phase). Wax increases tortuosity and supplies low-polarity domains; gum regulates water uptake; and glycerol acts as a buffer. Coordinated adjustment of these factors yields coatings that protect aroma and moisture simultaneously, an aspect seldom addressed for candelilla-based matrices. From a sensory standpoint, slowing hexanal diffusion would delay rancidity in fatty snacks, whereas limiting limonene loss would preserve the bouquet of citrus fruit [61,62]. In fresh dairy products, films rich in wax and gum could curb the leakage of floral volatiles such as linalool without hindering the release of desirable lactonic notes [63,64]. Mechanistically, two coupled phenomena explain our selectivity: (i) steric exclusion/free-volume control wax densification reduces accessible micro-voids and lowers D [38] and (ii) differential solubility hydrophobic wax domains decrease S for polar species while offering favorable partitioning for terpenes, which can simultaneously plasticize the matrix [31,53]. The net permeability P(D,S) thus shifts with RH and VOC identity; designing for specific aroma profiles requires placing the formulation at a point where terpene S is low enough (via wax continuity) to offset any plasticization-driven rise in D.

This study has limitations: assays were conducted at 25 °C and two relative-humidity settings, which do not encompass the full thermal and hygrometric range of the supply chain [65,66]. In particular, cold-chain temperatures (e.g., refrigeration) and near-saturation humidity conditions are expected to modify VOC sorption and diffusion (via moisture-driven plasticization and potential wax-phase reorganization) and were intentionally outside the scope of the present controlled screening. Long-term integrity under thermal cycling and pH variation must be assessed, along with the diffusion of other volatiles (acids, amines, and sulfur compounds) relevant to different foods. Incorporating antioxidants or antimicrobials could yield synergistic functions worthy of investigation [67,68,69], and industrial scale-up will require validating emulsion rheology in continuous processes and analyzing cost–benefit ratios versus conventional coatings. Limitations tied to temperature and RH are well documented: both moisture uptake and penetrant solubility increase with RH, which raises D through plasticization and can change S non-linearly [33]. Moreover, continuous wax layers are particularly sensitive to humidity-induced discontinuities and crystallinity changes that degrade the barrier [49]. Future tests should therefore report VOC permeation at multiple RH/T pairs and, for terpenes, quantify time-dependent plasticization to separate reversible sorption from irreversible morphological change.

In summary, the 0.4% wax/0.8% gum/0.2% glycerol formulation emerges as the most balanced, combining high density, low water permeability, and strong efficacy in retarding both aldehyde and terpene diffusion. The SPME-GC/MS methodology integrated with factorial design and linear quadratic kinetic modeling offers a useful screening framework for studying microstructure–selectivity relationships and informs subsequent, more comprehensive designs.

## 5. Conclusions

This screening study shows that candelilla wax/guar gum films can be engineered as dual barriers against water vapor and volatile aroma compounds and that the wax–gum–glycerol ratio controls both the magnitude and the temporal stability of transport. Using a 2^3−1^ fractional factorial design coupled with HS-SPME/GC–MS and linear–quadratic kinetic modeling allowed us to resolve main-effect trends with reproducible, batch-independent estimates of barrier performance.

Across the tested space, wax enrichment compacted the matrix, lowered WVTR/WVP, and attenuated the late-time acceleration observed in open hydrophilic films; EFD (0.4% wax, 0.8% gum, and 0.2% glycerol) delivered the best barrier–selectivity compromise, cutting the cumulative permeability of aldehydes and terpenes by up to 80% while maintaining kinetic stability over 12 h. Mechanistically, the data are consistent with steric/tortuosity increases and polarity-driven solubility differences that penalize terpenes in dense wax-laden domains and stabilize rates against penetrant-induced softening.

The results provide actionable criteria for edible-coating formulation: (i) maintain moderate-to-high wax to raise tortuosity and reduce plasticization; (ii) leverage gum content to densify the hydrophilic continuum and control water uptake; and (iii) keep glycerol at low levels to fine-tune free volume without over-softening. These guidelines support aroma retention while meeting moisture-barrier requirements in sustainable coatings.

Inferences are restricted to the main effects of a resolution-III fractional design (possible aliasing with two-factor interactions) and to assays at 25 °C under 33–67% RH; humidity/temperature dependencies and the time-dependent plasticization of terpenes should therefore be quantified explicitly in broader matrices and conditions.

Future work should expand the design with response surface optimization, additional volatile classes and multifunctional additives, and validation in real food systems, thereby translating the present screening into deployable bio-based packaging solutions.

## Figures and Tables

**Figure 1 foods-15-00233-f001:**
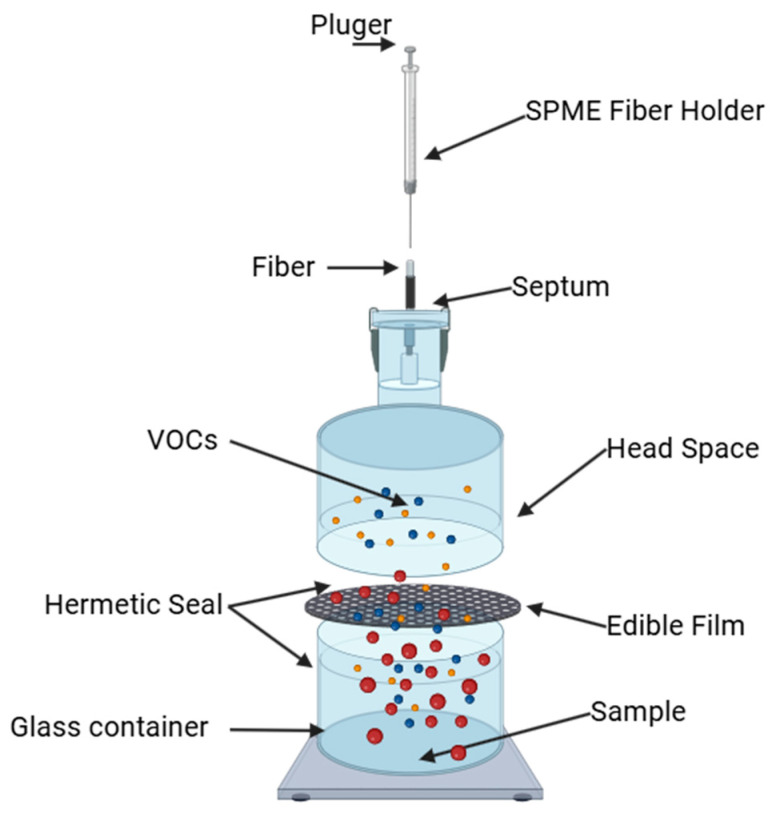
Two-chamber glass diffusion cell used to determine the permeability of edible films to volatile organic compounds. A defined quantity of VOCs is placed in the donor chamber; these molecules diffuse through the film into the headspace of the receiver chamber, where the analytes are adsorbed onto an SPME fiber and subsequently quantified by GC–MS.

**Figure 2 foods-15-00233-f002:**
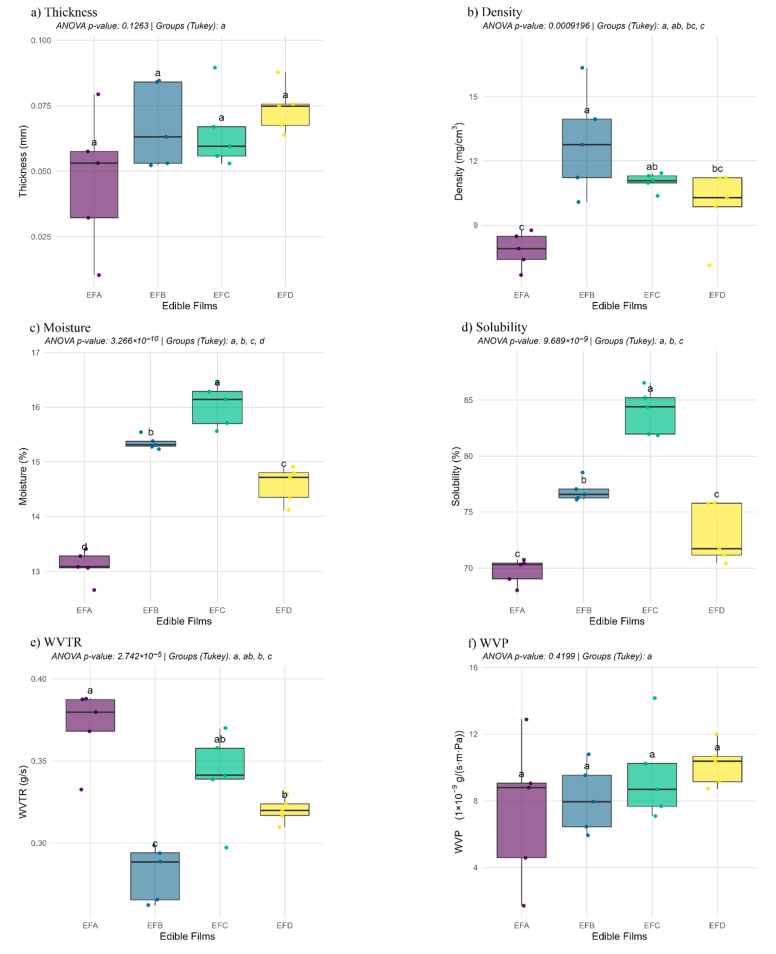
Attributes and water-barrier performance of candelilla wax edible films. Box-plot summaries of (**a**) film thickness, (**b**) apparent density, (**c**) moisture content, (**d**) aqueous solubility, (**e**) water vapor transmission rate (WVTR), and (**f**) water vapor permeability (WVP) for four formulations: EFA (0.4% guar gum:0.2% wax:0.2% glycerol), EFB (0.4%:0.4%:0.3%), EFC (0.8%:0.2%:0.3%), and EFD (0.8%:0.4%:0.2%). Different letters denote statistically significant differences among means.

**Figure 3 foods-15-00233-f003:**
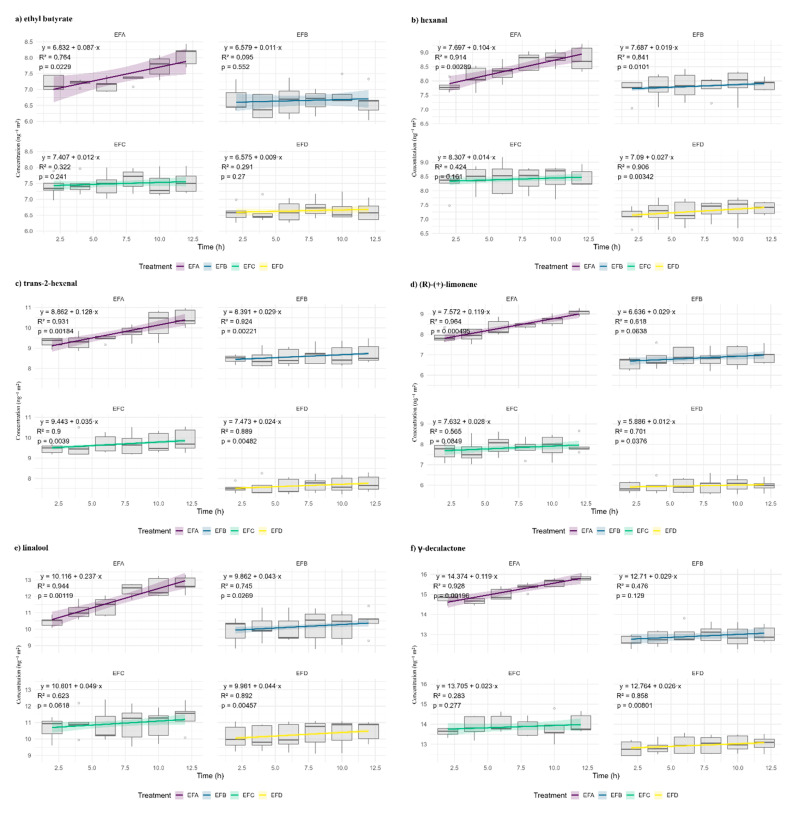
Volatile organic compound kinetics and linear model fit. Concentration profiles, C(t), for (**a**) ethyl butyrate, (**b**) hexanal, (**c**) trans-2-hexenal, (**d**) (R)-(+)-limonene, (**e**) linalool, and (**f**) γ-decalactone, permeating through four film formulations: EFA (0.4% gum:0.2% wax:0.2% glycerol; purple), EFB (0.4%:0.4%:0.3%; blue), EFC (0.8%:0.2%:0.3%; green), and EFD (0.8%:0.4%:0.2%; yellow). Solid lines represent the linear regression C(t) = β_0_ + β_1_t, while shaded bands denote 95% confidence intervals. Ordinates: ng of analyte recovered in the receiver chamber; abscissae: exposure time (h).

**Figure 4 foods-15-00233-f004:**
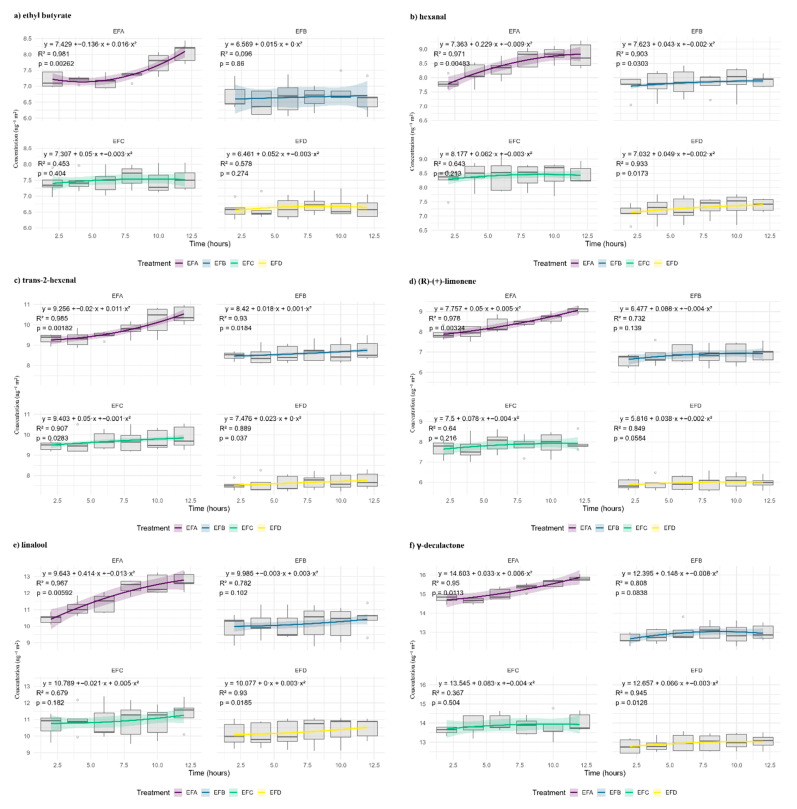
Volatile organic compound kinetics and quadratic model fit. Concentration profiles, C(t), for (**a**) ethyl butyrate, (**b**) hexanal, (**c**) trans-2-hexenal, (**d**) (R)-(+)-limonene, (**e**) linalool, and (**f**) γ-decalactone, permeating through four film formulations: EFA (0.4% gum:0.2% wax:0.2% glycerol; purple), EFB (0.4%:0.4%:0.3%; blue), EFC (0.8%:0.2%:0.3%; green), and EFD (0.8%:0.4%:0.2%; yellow). Solid lines represent the quadratic regression C(t) = β_0_ + β_1_t + β_2_t^2^, and shaded bands denote 95% confidence intervals. Ordinates: ng of analyte recovered in the receiver chamber; abscissae: exposure time (h).

**Figure 5 foods-15-00233-f005:**
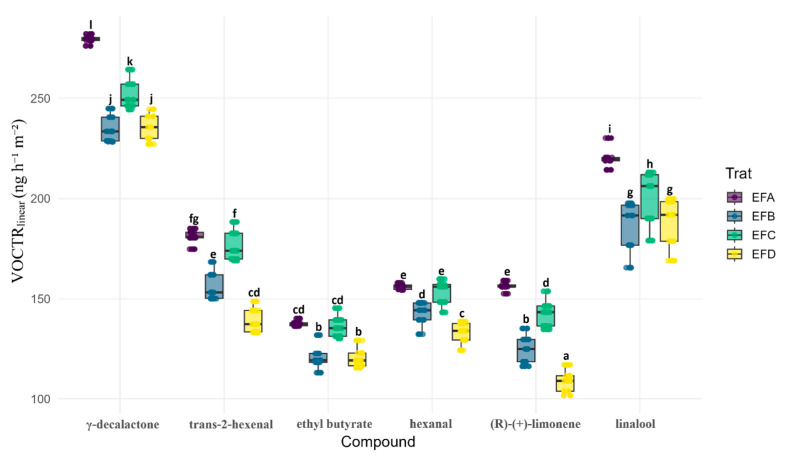
Linear volatile transmission rate (VOCTR_linear_) through candelilla wax films. Color key: EFA (0.4% gum:0.2% wax:0.2% glycerol), purple; EFB (0.4%:0.4%:0.3%), blue; EFC (0.8%:0.2%:0.3%), green; and EFD (0.8%:0.4%:0.2%), yellow. Center line = median; box = Q1–Q3; whiskers = 1.5 × IQR; and dots = individual slopes. Different letters over the boxes denote significant formulation effects within each metabolite (two-way ANOVA, Formulation × Metabolite, and N = 30; Sidak-adjusted post hoc, α = 0.05).

**Figure 6 foods-15-00233-f006:**
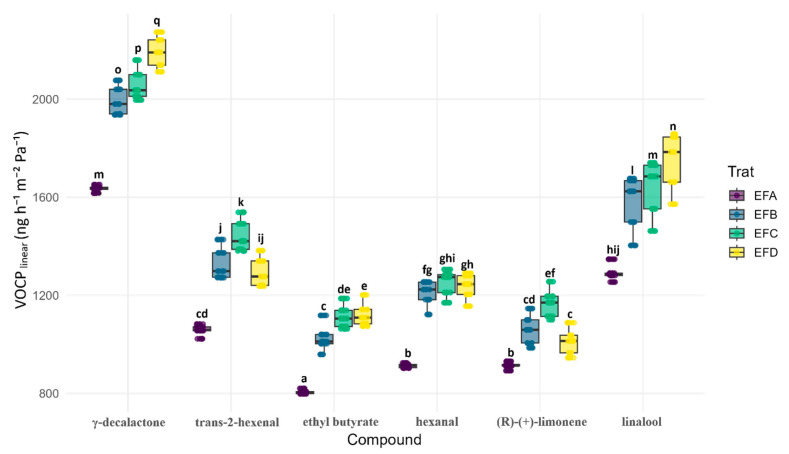
Linear volatile permeability coefficient (VOCP_linear_) through candelilla wax films. Color key: EFA (0.4% gum:0.2% wax:0.2% glycerol), purple; EFB (0.4%:0.4%:0.3%), blue; EFC (0.8%:0.2%:0.3%), green; and EFD (0.8%:0.4%:0.2%), yellow. Center line = median; box = Q1–Q3; whiskers = 1.5 × IQR; and dots = individual slopes. Different letters over the boxes denote significant formulation effects within each metabolite (two-way ANOVA, Formulation × Metabolite, and N = 30; Sidak-adjusted post hoc, α = 0.05).

**Figure 7 foods-15-00233-f007:**
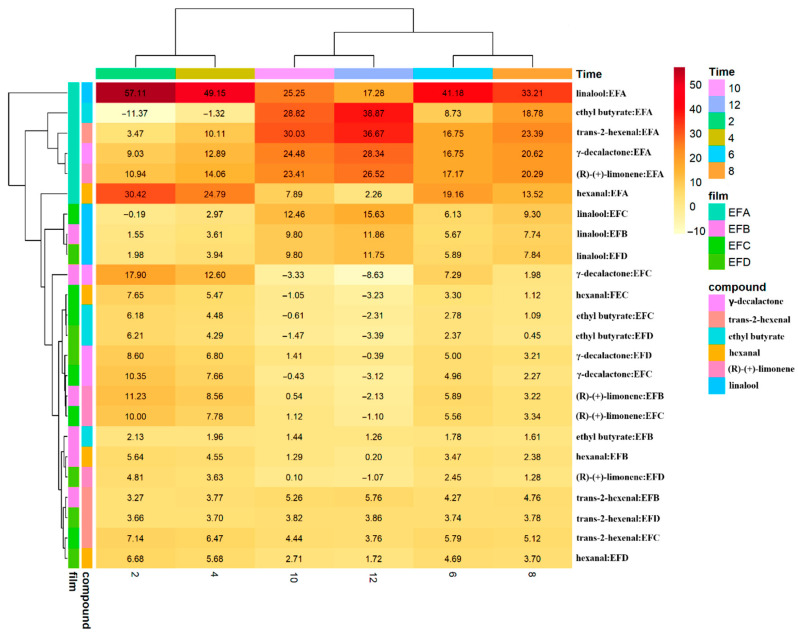
Linear volatile transmission rate (VOCTR_quadratic_) through candelilla wax films. The heat map displays mean VOCTR_quadratic_ values for every formulation–analyte pair at 2, 4, 6, 8, 10, and 12 h. Numerical entries are arithmetic means of three independent batches. Hierarchical clustering (Euclidean distance, average linkage) groups rows (formulations) and columns (analytes) by kinetic similarity. Formulations: EFA (0.4% gum:0.2% wax:0.2% glycerol); EFB (0.4%:0.4%:0.3%); EFC (0.8%:0.2%:0.3%); and EFD (0.8%:0.4%:0.2%).

**Figure 8 foods-15-00233-f008:**
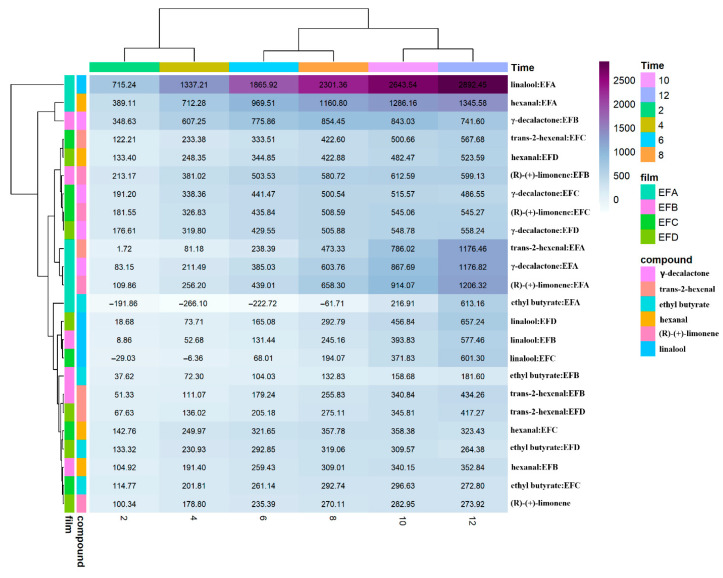
Quadratic volatile permeability coefficient (VOCP_quadratic_) through candelilla wax films. The heat map displays mean VOCP_quadratic_ values for every formulation–analyte pair at 2, 4, 6, 8, 10, and 12 h. Numerical entries are arithmetic means of three independent batches. Hierarchical clustering (Euclidean distance, average linkage) groups rows (formulations) and columns (analytes) by kinetic similarity. Formulations: EFA (0.4% gum:0.2% wax:0.2% glycerol); EFB (0.4%:0.4%:0.3%); EFC (0.8%:0.2%:0.3%); and EFD (0.8%:0.4%:0.2%).

**Figure 9 foods-15-00233-f009:**
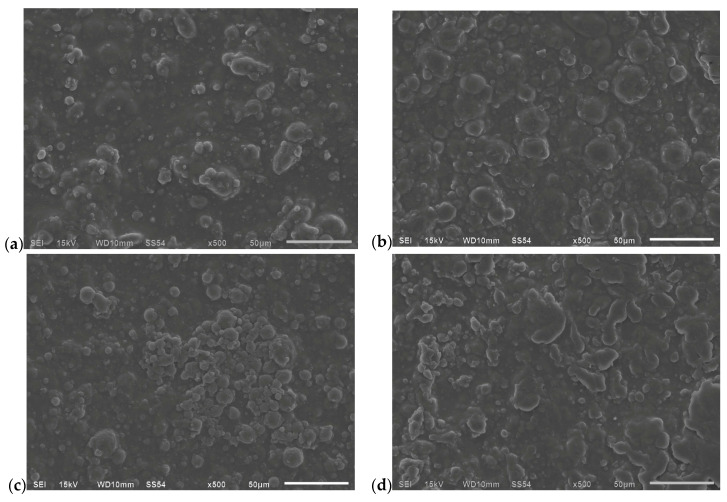
SEM surface micrographs of the edible films (SEI; HV 15 kV; WD 10 mm; spot 54; ×500; and 50 μm scale bar). Panels (**a**–**d**) correspond to formulations EFA, EFB, EFC, and EFD, respectively. All micrographs were acquired under identical SEM settings and are provided for qualitative comparison of surface morphology (domain density, size distribution, and degree of coalescence) across formulations.

## Data Availability

The original contributions presented in this study are included in the article. Further inquiries can be directed to the corresponding author.

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
