# Peer review of "Selective Permeability of Volatile Organic Compounds in Candelilla Wax Edible Films"

_foods, 2026, doi:10.3390/foods15020233_

Round 1

Reviewer 1 Report

Comments and Suggestions for Authors

This study investigated the permeability of VOCs through edible films made of candelilla wax and guar gum, offering new insights into their role as aroma and moisture barriers. This paper provided significant support for the development of edible packaging films for food. However, there are still many issues in the paper that need to be revised and improved. The specific points are as follows:

1.The experimental design of this paper has certain limitations. For instance, in terms of experimental conditions, the authors only studied the conditions of 25℃ and 33-67% relative humidity. This condition does not fully cover the range of food transportation and storage (such as refrigeration or high humidity conditions).

2.The second issue of this paper is that it only detected six volatile components, but failed to analyze other common volatile components in fresh food, such as acids and amines.

3.The characterization analysis of these edible films was relatively weak; they only conducted SEM analysis. The structural analysis data of this paper seem to provide relatively insufficient support for the interpretation of the relationship between the microstructure and the penetration of VOCs.

4.Furthermore, the authors did not conduct coating verification for specific fresh food (such as fruits, vegetables, fresh meat, etc.).

5.In the experimental design, the author did not make a blank comparison with the VOCs permeation data of existing commercial edible membranes (such as chitosan, etc.). As a result, it is difficult to compare the performance advantages of the membrane prepared by the author.

6.Introduction: It is suggested that the author should appropriately add some textual descriptions about the research progress related to VOCs in edible films.

7.Line 93-105: Why did you choose this factor design? How can this design analyze the interaction effects between various factors? How can this design provide a basis for optimizing the formula?

8.Line 178, 181: 2.6.1, 2.6.2: The suggested abbreviations for the titles should be consistent with the initial letters of the words in the titles.

9.Figure 2: The explanatory text for the significance analysis in the figure is suggested to be deleted, as it has already been covered in Section 2.12.

10.Fig 3, 4: The author only examined the data for 12 hours. Personally, I feel that 12 hours is rather short and it differs significantly from the actual shelf life of the food.

11.Fig 9: The conditions of the four graphs in Figure 9 need to be further explained.

Reviewer 2 Report

Comments and Suggestions for Authors

The manuscript is generally well-written and scientifically sound, a few ascpects may be addressed to improve this manuscript.

While the six VOCs are mentioned as key compounds to food systems, the reasons for their selection should be elaborated further.

There are inconsistencies in the abbreviations used for permeability parameters. In Equations 8–11 and Figures 5–8, terms such as  VOCPquadratic,  VOCTRquadratic  etc., appear. Please ensure uniform formatting throughout the text, equations, and captions. Line 542 "COVP" is likely a typographical error, please check through the whole manuscript.

It is suggested to describe how specific SEM structures may influence diffusion pathways for VOCs.

Reviewer 3 Report

Comments and Suggestions for Authors

The authors present a detailed study of edible films made from a combination of waxes, guar gum and glycerol and their permeability with different aroma stimulants. The authors have explained in great detail all the analytical steps involved in performing the experiments, making them easier to replicate. However, while technically the work is well-written, there are several improvement opportunities. Firstly, why exactly was this wax chosen (is it scalable, is it really more sustainable), and what is the combination? The introduction lacks the backstory of why each component was used and what other authors have already achieved in this field. The second opportunity is the lack of design of the experiment setup with different combinations - how and why the authors stick with this exact combination of the components? With the use of the DoF we would know exactly which component influences how the edible films. As basically the aromas are going through adsorption and diffusion through pores, some kind of porosimetry measurement would be more useful to find the numbers or volume of the pores (rather than using SEM).

Round 2

Reviewer 1 Report

Comments and Suggestions for Authors

The authors have thoroughly revised the manuscript, which is now recommended for acceptance.